biophysics/computational biology/behaviour

flocking, topological and metric interaction, mathematical modelling, statistical analysis, swarm behaviour

**Author for correspondence:**
Rumi De
e-mail: rumi.de@iiserkol.ac.in

# Efficient flocking: metric versus topological interactions

## Vijay Kumar[1,2] and Rumi De[1]

[1]Department of Physical Sciences, Indian Institute of Science Education and Research Kolkata, Mohanpur 741246, West Bengal, India
[2]Centre for Computational and Data-Intensive Science and Engineering, Skolkovo Institute of Science and Technology, Nobelya Ulitsa 3, Moscow, 121205, Russia

VK, 0000-0002-7402-5193; RD, 0000-0002-8831-5829

Flocking is a fascinating phenomenon observed across a wide range of living organisms. We investigate, based on a simple self-propelled particle model, how the emergence of ordered motion in a collectively moving group is influenced by the local rules of interactions among the individuals, namely, metric versus topological interactions as debated in the current literature. In the case of the metric ruling, the individuals interact with the neighbours within a certain metric distance; by contrast, in the topological ruling, interaction is confined within a number of fixed nearest neighbours. Here, we explore how the range of interaction versus the number of fixed interacting neighbours affects the dynamics of flocking in an unbounded space, as observed in natural scenarios. Our study reveals the existence of a certain threshold value of the interaction radius in the case of metric ruling and a threshold number of interacting neighbours for the topological ruling to reach an ordered state. Interestingly, our analysis shows that topological interaction is more effective in bringing the order in the group, as observed in field studies. We further compare how the nature of the interactions affects the dynamics for various sizes and speeds of the flock.

## 1. Introduction

Cohesive group formation is one of the most eye-catching displays in nature. It is observed among various species, such as, flock of birds [1,2], school of fishes [3], swarm of prey [4,5], colony of bacteria [6], aggregation of cells [7] and pedestrian crowd [8]. The instances of collective behaviours have also been demonstrated in non-living systems including multi-agent robots [9], vibrated discs [10], artificial microswimmers [11] etc. Until now, a great effort has been devoted to understanding the underlying universal mechanisms of such spontaneous emergence of ordered motion irrespective of the nature of the constituent entities. As a result, several theoretical approaches have been developed to study the

collective dynamics [1,2,12–19]. Among these, the self-propelled particle-based model, since the seminal work of Vicsek [20], remains one of the most favourite choices as it could replicate both system-specific and universal features of the collective behaviours observed across a wide range of species. Moreover, because of its simplicity, one could easily test the model against the experimental observations. Several studies have revealed that collective motion can emerge from simple local rules of interaction among the individuals without a leader or any kind of a central control [21]. The most commonly discussed interactions include short-range repulsions and long-range attractions among the individuals, or the alignment of velocities along with the nearest neighbours [1,22]. Many intriguing questions, thus arise, e.g. how the individuals keep track of its neighbour in a large extended group, how far its range of interaction extends, or how many interacting neighbours do they require to establish such a coordinated motion of the whole group to flock together.

Following Vicsek's work, most of the self-propelled particle models have incorporated metric-based interaction rules according to which each individual interacts with its surrounding neighbours up to a certain metric distance. Hemelrijk and Hildenbrandt have demonstrated that metric-based interactions incorporating cohesion, alignment and separation rules can explain the shapes and patterns of fish schooling [23]. Using a simple model with metric rules, Couzin *et al.* have shown how efficient information transfer and decision-making can occur in animal groups [24]. Another self-propelled particle model introduced by Bhattacharya and Vicsek indicates the mechanisms of the synchronized landing of a flock of birds performing foraging flights [25]. There are also other metric-based interaction models that have given many insights into the orientational order, cluster formations, synchronization and spatial sorting in collective groups [26–28].

However, in contrast to the metric-based interaction rules, it has recently been suggested in an experimental finding that the birds in a flock tend to interact with a fixed number of nearest neighbours [29]; this has been termed as topological interaction in the subsequent literature. Ballerini *et al.* have analysed the trajectories of flocks of a few thousand starlings and have shown that each bird interacts with a fixed number of neighbours (on an average six to seven neighbours) irrespective of their metric distance [29]. Camperi *et al.* based on a self-propelled particles model have shown that the topological models are more stable than the metric ones, and maximal stability is attained when topological neighbours are distributed evenly around each individual, i.e. the neighbours are chosen from a spatially balanced neighbourhood [30]. Further, using network and graph-theoretic approaches coupled with a dynamical model, Shang and Bouffanais have studied the consensus reaching process with topologically interacting self-propelled particles. They have shown regardless of the group size, a value of close to 10 neighbours speeds up the rate of convergence to the consensus to an optimal level where all particles interact with the entire group [31].

Several studies on metric interactions and quite a few on topological interactions have been carried out; however, what kind of inter-agents interaction governs the macroscopic collective order among diverse species is not yet well-understood [32,33]. On the one hand, animals can estimate absolute distance by various methods, including retinal image size and different motional cues [34]. Therefore, estimating the metric distance among the surrounding neighbours could be a natural interaction. On the other hand, the sensory and cognitive limitation of an individual indicates that instead of interacting with all members, topological interaction with a few neighbours could play an important role in establishing collective order in the group. In this paper, based on a simple self-propelled particle model, we investigate how the metric versus topological interactions affect the emergence of ordered motion in a flock. We study the dynamics of flocking by both varying the range of interaction radius and the number of interacting topological neighbours among the individuals. Our theory predicts the existence of a certain threshold interaction radius for the metric ruling and a threshold number of interacting neighbours for the topological ruling for the whole group to reach an ordered state. We further explore how the nature of interactions influences the overall dynamics for different speeds and group sizes of the flock. It shows that the lower speed is more beneficial for the group to flock together compared with the higher speed. Furthermore, in the case of metric interaction, the threshold value decreases with an increase in flock sizes irrespective of the flock speeds. On the contrary, in topological interaction, increasing the group size increases the threshold value for flocks moving with the higher speed; however, it remains unaffected for flocks with a lower speed. Overall, the topological interaction turns out to be beneficial and effective in bringing order in the flock.

## 2. Theoretical model

We consider a group of $N$ active particles in two-dimensional space where each particle is characterized by its position, $\mathbf{r}_i$, and the velocity, $\mathbf{v}_i = v_0 \cos \theta_i \hat{x} + v_0 \sin \theta_i \hat{y}$, where $v_0$ is the magnitude and $\theta_i$ is the

direction of the velocity of the $i$th particle. To mimic the real-life scenario in the field, we consider that the particles move in open space (hence, the boundary is not restricted). Moreover, the particle velocity can change in magnitude as well as in direction depending on the local inter-particle interactions. In a simple way to incorporate this feature, we consider a velocity alignment interaction between the particles and the surrounding neighbours. In the case where the interaction is governed by the metric distance, each particle interacts with its neighbours within a certain reaction radius, $r_{int}$. On the other hand, in the case of topological interaction, each particle chooses to interact with a certain $N_r$ number of nearest neighbours. Thus, the equation of motion is given by,

$$\frac{d\mathbf{v}_i}{dt} = \frac{\alpha}{N_{int}} \sum_{k=1}^{N_{int}} (\mathbf{v}_k - \mathbf{v}_i) - \gamma \mathbf{v}_i + \boldsymbol{\xi}_i, \tag{2.1}$$

where $\mathbf{v}_i$ is the velocity of the $i$th particle. Here, the first term on the right-hand side denotes the velocity alignment interaction of the $i$th particle with the surrounding neighbours. The summation is performed over $N_{int}$ number of neighbours interacting with the $i$th particle following the metric or the topological rule of interactions. The motion of the surrounding neighbours influences the velocity of an individual particle, and it responds by changing its velocity to equal the velocity of the neighbouring particles. Here, $\alpha$ represents the strength of the inter-particle interactions, $\gamma$ denotes the coefficient of the viscous drag experienced by the particle, and $\xi$ represents the noise resulting from the surrounding medium. However, since our main focus is to investigate how the nature of the interactions, be it metric or topology, influences the flock, here we concentrate on the simplest case where the damping term and the noise are negligible and thus not considered. We study the dynamics in dimensionless units. Time variable is scaled as $T = t/\tau$, the dimensionless velocity is given by $\mathbf{V} = \mathbf{v}/v_s$, the dimensionless speed is $V_0 = v_0/v_s$ where $v_s = l_s/\tau$; and other scaled parameters are interaction radius, $R = r_{int}/l_s$ and $\alpha_s = \alpha\tau$; where the constants $l_s$ and $\tau$ represent the characteristic length and time scale of the system.

# 3. Results

We study the dynamics of flocking by varying the range of the interaction radius, $R$, in the case of metric and the number of interacting neighbours, $N_r$, for the topological interactions. In our simulations, we consider a group of $N$ particles initially positioned randomly in a square box of size $L$. The initial directions of the velocity of the particles are chosen uniformly between 0 to $2\pi$ with a velocity magnitude, $V_0$, and then the positions and velocities of the particles evolve in time depending on the interaction with their neighbours following equation (2.1). We have investigated the dynamics for a wide range of parameter values; here, we present the results for some representative values by keeping the initial box size, $L = 25$, $\alpha_s = 1$, and the flock size, $N = 100, 200, 300, 500$. We have also studied the effect of varying initial velocity magnitude, $V_0$, on the flocking state; the results presented here are for two different regimes, at a lower speed, $V_0 = 0.01$, and at a higher speed, $V_0 = 1.0$, of the flock. The results are obtained, averaging over 100 such simulation trajectories starting from different initial configurations. It is worth noting that the group size, $N$, is motivated by the literature study of the size of flocks, schools and herds of various species ([24,35] and references therein). Many groups could also contain a large number of individuals; however, even within this range, our analysis clearly demonstrates the influence of variation in group sizes in the case of both metric and topological interactions.

Now, to characterize the ordered state, i.e. the flocking state when all individuals move along the same direction in unison, we consider the order parameter, $\phi$, to describe the degree of the collective order as the absolute value of the averaged normalized velocity of the group given by [20],

$$\phi = \frac{1}{N} \left| \sum_{i=1}^{N} \frac{\mathbf{V}_i}{|\mathbf{V}_i|} \right|, \tag{3.1}$$

where $N$ is the total number of particles in the group, $\mathbf{V}_i$ is the velocity of the $i$th particle, and $|\mathbf{V}_i|$ is the magnitude, i.e. the speed of the particle. The order parameter, $\phi$, measures the global order in the system at any given time instant, $T$. $\phi$ approaches unity for ordered motion and zero for disordered state.

We further investigate how the order or the degree of similarity varies spatially in time by analysing the spatial correlation function, $C(r)$,

$$C(r) = \frac{\sum_{i,j}^{N} \widehat{\mathbf{V}}_i \cdot \widehat{\mathbf{V}}_j \delta(r - r_{ij})}{\sum_{i,j}^{N} \delta(r - r_{ij})}, \tag{3.2}$$

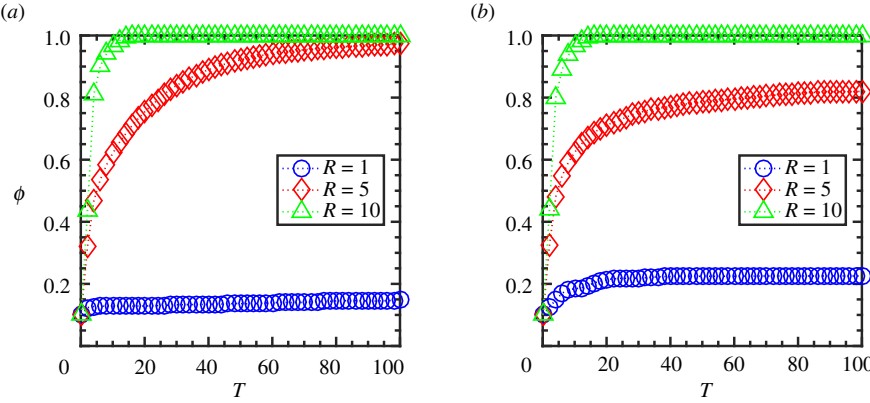

**Figure 1.** Metric interaction: order parameter, $\phi$, as function of time, $T$, for different interaction radius ($R = 1$, 5, 10) and initial flock speed, $V_0$, (a) at a lower speed ($V_0 = 0.01$) and (b) at a higher speed ($V_0 = 1.0$). (The flock size is $N = 100$.)

where $C(r)$ measures the velocity correlation between a pair of particles, $i$ and $j$, at a distance $r$ at a given time $T$. Here, $\widehat{\mathbf{V}}_i = \mathbf{V}_i/|\mathbf{V}_i|$ is the unit velocity vector of the $i$th particle and $r_{ij} = |\mathbf{r}_i - \mathbf{r}_j|$ denotes the distance between a pair of particles, $i$ and $j$. The Kronecker delta, $\delta(r - r_{ij}) = 1$ if $r = r_{ij}$, and $\delta(r - r_{ij}) = 0$ if $r \neq r_{ij}$. To calculate the spatial average, summation is performed over all pairs with a distance $r_{ij}$ between $r$ and $r + dr$ and then it is divided by the number of such pairs (the number is given by the denominator term). This gives a fair estimate of the velocity correlation, $C(r)$, between two individuals in a group [2].

## 3.1. Metric interaction: influence on flocking state

Figure 1a,b shows the time evolution of the order parameter, $\phi$, for different interaction radius, $R$, at a lower and a higher flock speed, $V_0$, keeping the flock size, $N = 100$, for a representative simulation trajectory. As seen from the figure, the order parameter increases with an increase in the range of interaction since the agent-particles get to interact with more neighbours with an increasing radius, which facilitates to bring in higher order in the group. Furthermore, it shows that for a given interaction radius, $R$, the group moving with a lower speed, $V_0 = 0.01$, results in more order in the system (figure 1a) compared with the higher speed, $V_0 = 1.0$ (figure 1b). It is because the particles with a low speed move with the same surrounding neighbours for a considerable time; thus, they get sufficient time to align their velocities. Whereas the particles moving at high speed go through rapid changes in their surrounding neighbours; thus, the overall order in the group decreases.

We further investigate the spatial ordering in the group while it approaches the flocking state by analysing the velocity correlation function, $C(r)$, at different time instances, $T$, for two reaction radii, $R = 5$ and 10 shown in figure 2a–d. We calculate the spatial statistical average by summing over all pairs of particles in the group as described by equation (3.2) for one representative simulation trajectory. Initially, since the particles are positioned randomly, in the beginning, at $T = 1$, the velocity correlation function, $C(r)$, is close to zero over the entire spatial extent $r$ of the system as could be seen from figure 2a,b. As time evolves, at $T = 5$, the particles located within the interaction radius, $R = 5$, start interacting and gradually tend to align along with their neighbours. Thus, the correlation between the particle velocities increases at a shorter distance $r$; however, the correlation function stays close to zero between the particles at a large distance as the motion remains uncorrelated. As time progresses further (shown at $T = 1000$), the correlation functions, $C(r)$, approaches unity, and the entire group reaches an ordered state. It could be seen from the figure that for a given flock speed, the agent-particles with a higher interaction radius (figure 2c,d) reach the flocking state faster compared with a smaller radius (figure 2a,b). Similarly, if we compare the speeds, for a smaller range of interaction, $R = 5$, the group with a lower speed tends to approach the ordered state faster compared with the group with a higher speed, as shown in figure 2a,b. However, for a higher interaction radius, ($R = 10$), the variation in flock speeds does not significantly alter the dynamics, as could be seen in figure 2c,d.

To further explore, we study the order parameter, $\phi$, at steady state as a function of interaction radius, $R$, for different group size, $N$, and speed, $V_0$, of the flock, shown in figure 3a,b (values are obtained averaging over hundred simulation trajectories). We find that a certain threshold interaction radius, $R_{\text{th}}$, is required for the group to reach the ordered state ($\phi \sim 1$), i.e. for the group to flock together.

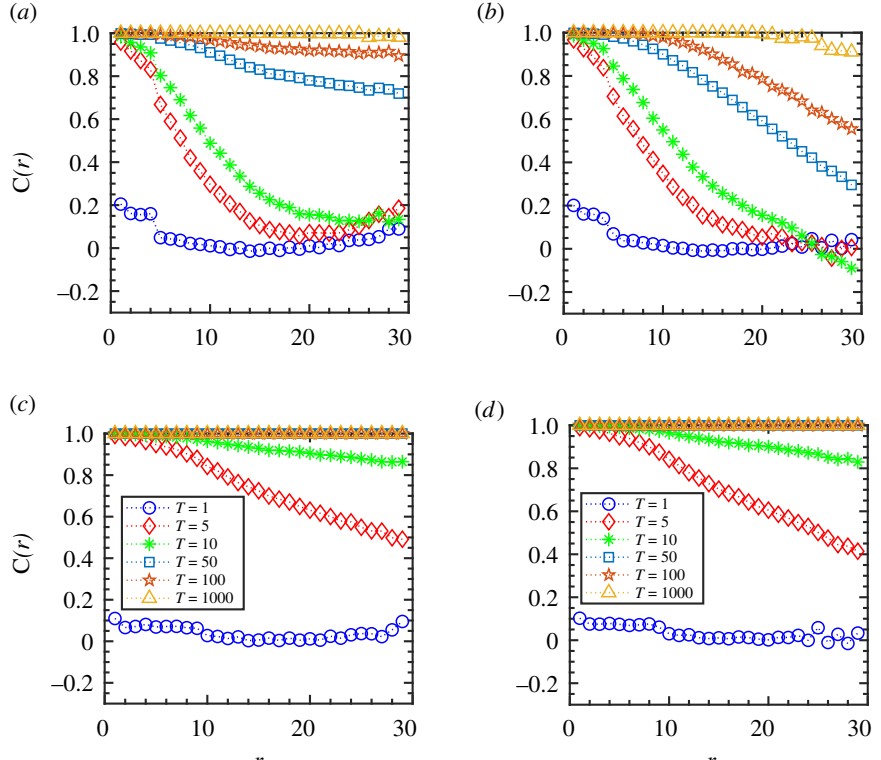

**Figure 2.** Metric interaction: velocity correlation function, $C(r)$, at different time instances, $T$, for two interaction radii, $R$, and initial flock speed, $V_0$, where ($a$) $R = 5$ and $V_0 = 0.01$, ($b$) $R = 5$ and $V_0 = 1.0$, ($c$) $R = 10$ and $V_0 = 0.01$ and ($d$) $R = 10$ and $V_0 = 1.0$. The flock size is $N = 100$. (Figure legends of ($a$,$b$) are also the same as in ($c$,$d$) indicating different time instances, $T$.)

Moreover, the threshold value of the interaction radius, $R_{th}$, decreases with the increase in the flock size, $N$. With increasing group size, the individuals get surrounded by more closely spaced interacting neighbours. As the average distance between the particles decreases, the threshold value of interaction, $R_{th}$ also decreases. On the other hand, for a given flock size, the threshold value, $R_{th}$, increases with an increase in the flock speed, as shown in figure 3$a$,$b$. Thus, the lower speed turns out to be beneficial in establishing the order in the group as the individuals moving with low speed get to spend considerable time with the same neighbours to align their velocities and thus, require to interact comparatively with a smaller number of particles and at a shorter distance for the whole group to reach the flocking state (values are presented in table S1 in electronic supplementary material).

## 3.2. Topological interaction: influence on flocking state

We now investigate how the flocking state is affected when the individuals interact topologically, i.e. when they interact with a certain number of nearest neighbours irrespective of their metric distances. Figure 4$a$,$b$ shows the steady-state value of the order parameter, $\phi$, as a function of the number of nearest neighbours, $N_r$, for different flock size, $N$, and initial speed, $V_0$ (averaging has been done over 100 simulation trajectories). Our analysis shows that also in the case of topological interaction, a threshold number of nearest neighbours, $N_{th}$, is required for the whole group to attain the ordered state ($\phi \sim 1$). Moreover, for a given flock size, $N$, if we increase the speed from $V_0 = 0.01$ (figure 4$a$) to a higher value, $V_0 = 1.0$ (figure 4$b$), the threshold value, $N_{th}$, increases. Interestingly, in contrast to the metric ruling, here, the variation in flock sizes does not alter the threshold number for flocks moving with a lower speed (figure 4$a$). On the other hand, for the high-speed case, increasing the flock size increases the threshold value to reach the ordered state (figure 4$b$). Furthermore, we have studied the time evolution of the order parameter, $\phi$, and the spatial velocity correlation, $C(r)$, for the topological interactions which have been presented in the electronic supplementary material. It is noteworthy that other studies have also observed that the cohesion of the whole group moving with a constant speed increases with an increase in the topologically interacting neighbours. The threshold or the optimal

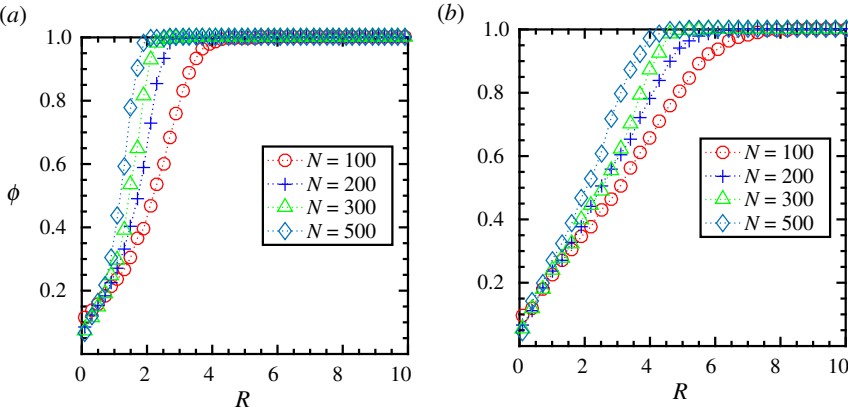

**Figure 3.** Metric interaction: the order parameter, $\phi$, at steady state as a function of interaction radius, $R$, for different flock size, $N = 100$, 200, 300, 500 and initial speed, $V_0$, (a) at a lower speed ($V_0 = 0.01$) and (b) at a higher speed ($V_0 = 1.0$).

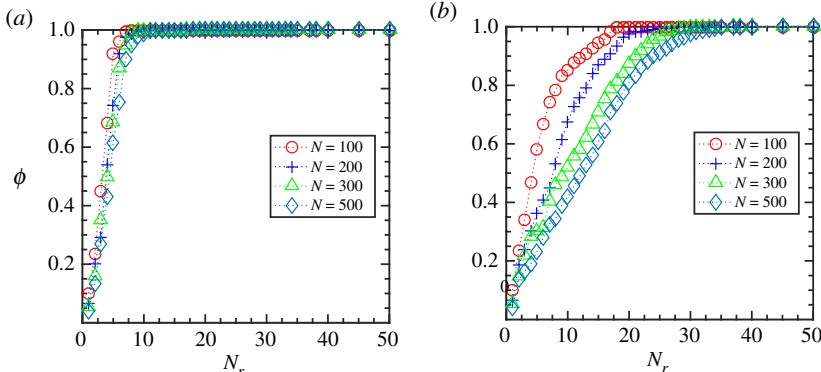

**Figure 4.** Topological interaction: the order parameter, $\phi$, at steady state as a function of number of interacting topological neighbours, $N_r$, for varying flock sizes, $N = 100$, 200, 300, 500 and initial flock speed, $V_0$, (a) at a lower speed ($V_0 = 0.01$) and (b) at a higher speed ($V_0 = 1.0$).

number of neighbours is independent of the swarm size, as we find in the case of flocks moving at a slower speed [31].

## 3.3. Metric versus topological interactions

So far, we have studied the influence of metric and topological interactions on the emergence of the ordered state. Here, we analyse which of these local interactions is favourable for the group to flock together. For a given number of topologically interacting neighbours $N_r$, we calculate the mean distance, $R_m$, among the interacting agent-particles in the steady state (averaging has been done over 100 simulation trajectories). Figure 5a,b shows the mean distance, $R_m$, as a function of interacting neighbour, $N_r$, for different flock sizes as well as speeds. Interestingly, the mean emergent distance, $R_m$, when plotted against $\frac{N_r}{N}$, i.e. in terms of the fraction of total group member, turns out to be invariant to the change in flock size, $N$, as could be seen in figure 5c,d. Similar results are also obtained for the metric interaction where for a given interaction radius, $R$, the mean number of interacting particles, $N_m$, has been calculated (plots are shown in the electronic supplementary material).

Now in order to compare how the metric and the topological ruling could fare in establishing the order in the flocking state, we evaluate the minimal interaction distance and the minimal number of interacting agents required to reach the ordered state, i.e. when $\phi$ reaches unity. We define an efficiency parameter, $\eta = R_{th}N_m$, for the metric interaction and for the topological interaction, $\eta = N_{th}R_m$; where $R_{th}$ is the threshold value of the interaction radius required for the metric ruling and the threshold number of neighbours, $N_{th}$, for the topological ruling to attain the order state. (The values of $R_{th}$ and $N_{th}$ are obtained from figures 3 and 4 and the corresponding values of the mean emerging interacting neighbours, $N_m$, and the mean emerging interaction distance, $R_m$, are obtained from electronic supplementary material, figure S3, and figure 5. The

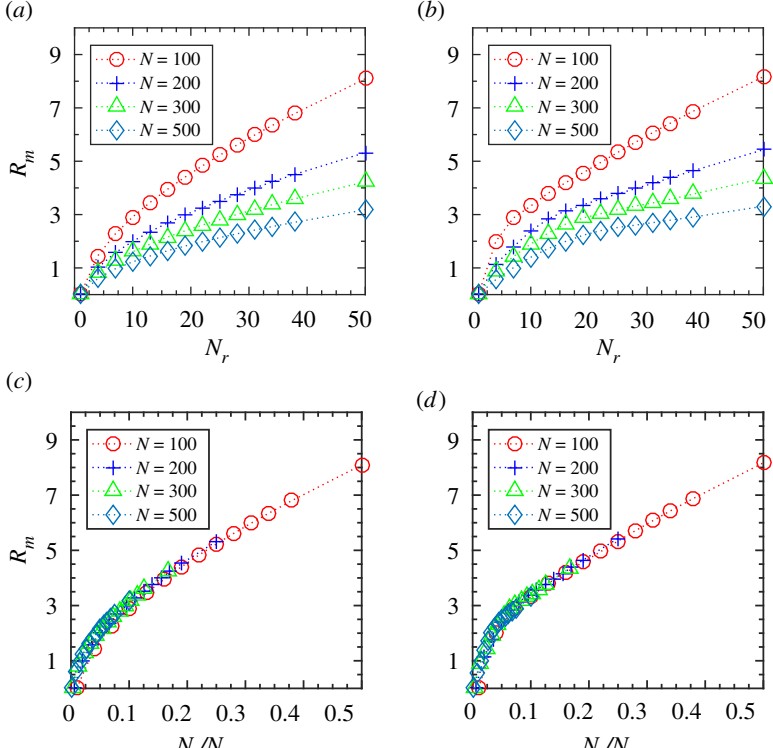

**Figure 5.** Variation of mean interaction distance, $R_m$, as a function of number of interacting topological neighbours, $N_r$, for varying flock sizes, $N = 100, 200, 300, 500$ and initial flock speed, $V_0$, (a) at a lower speed ($V_0 = 0.01$) and (b) at a higher speed ($V_0 = 1.0$). (c) Variation of $R_m$ as a function of $N_r/N$ for a lower speed ($V_0 = 0.01$) and (d) a higher speed ($V_0 = 1.0$).

values are presented in table S1 in the electronic supplementary material.) Therefore, comparatively, flocking is considered to be more efficient if the whole group reaches the ordered state with a lower value of $\eta$. For the quantitative analysis of the dynamics, we compare the bar plot of $\eta$ with varying flock sizes ($N = 100, 200, 300$ and 500) and speeds ($V_0 = 0.01$ and $V_0 = 1.0$) for both the metric and the topological interactions. As shown in figure 6a,b, with a decrease in the flock size, the difference between the metric and the topological interaction becomes more prominent, and the topological ruling becomes significantly efficient as compared with the metric ruling. Moreover, the values of $\eta$ are lower in the case of topological ruling for both low- and high-speed flocks. Furthermore, if we scale $\eta$ by the flock size $N$, as shown in figure 6c,d, topological interaction turns out to be a clear winner for smaller group size. Furthermore, the difference between the metric and topological interaction becomes more distinct for the higher speed compared with the flock moving with a lower speed.

## 4. Discussion

We have investigated, based on a simple self-propelled particle model, how the local rules of metric and topological interactions affect the dynamics of flocking. We have studied the emergence of order by varying the interaction distance and also the number of interacting nearest neighbours for various flock sizes and speeds. It is observed that in the case of the metric ruling, a certain threshold value of interaction radius is required to reach the ordered state. Similarly, for the topological ruling, a threshold value of the number of interacting neighbours is needed for flocking. Our study shows that the threshold value for a given flock size gets lowered for the group moving with a lower speed for both the metric and the topological interactions. Thus, the lower speed turns out to be beneficial since the individuals are required to interact comparatively with a smaller number of neighbours and at a shorter distance for the group to flock together. Furthermore, in metric interaction, increasing the group size decreases the threshold interaction radius required for flocking irrespective of the flock speeds. On the contrary, in the topological ruling, the variation in the group size does not affect the flocking dynamics in the case of lower speed. Our study shows in the case of flocks moving at a lower speed, the optimal number of topologically interacting neighbours is approximately 7–10 to

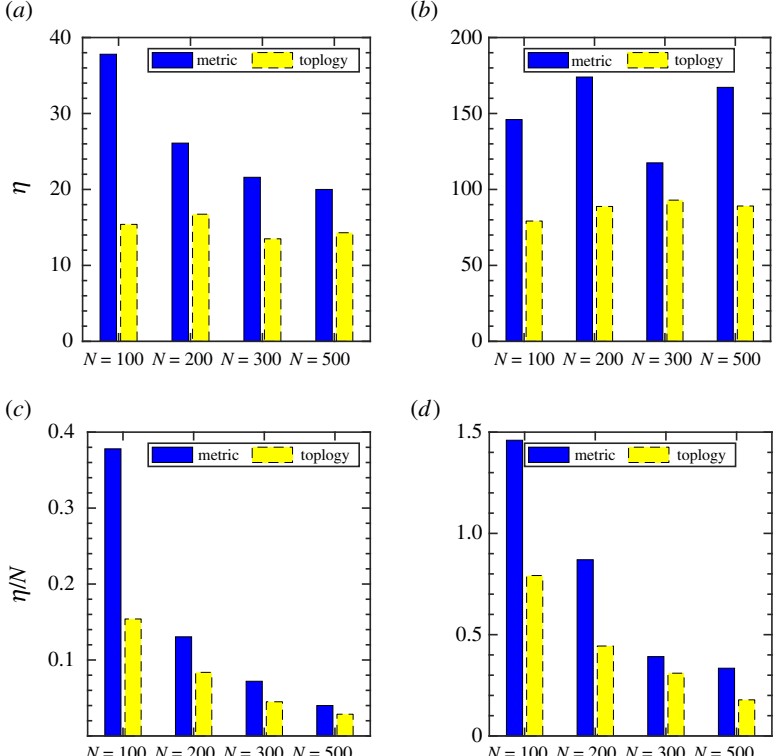

**Figure 6.** Metric versus topological interaction: (a) Bar plot of the flocking efficiency, $\eta$, for varying flock size, N, with a lower speed $(V_0 = 0.01)$ and (b) with a higher speed $(V_0 = 1.0)$. (c) Bar plot for the normalized efficiency $(\eta/N)$ for a lower flock speed $(V_0 = 0.01)$ and (d) for a higher speed $(V_0 = 1.0)$. Dark coloured (blue) bar represents metric interaction and light coloured (yellow) bar topological interaction.

achieve the ordered motion of the whole group, which has also been observed in other studies [29,31]; moreover, the number does not depend on the flock size as found in [31,36]. However, our analysis indicates that for groups moving at high speed, the optimal number of interacting neighbours require to be much higher to reach an ordered state, and also, the threshold number of interacting neighbours increases with the increase in the flock size, as shown in table S1 in the electronic supplementary material. Overall, the topological interaction turns out to be more efficient as compared with the metric interaction for the whole group to reach the order state. Moreover, the difference between the metric and the topological ruling becomes more pronounced with a decrease in the flock size. Furthermore, it is worth noting that apart from these two broad groups of local interactions—metric and topological rules—there are also studies considering a hybrid model of metric–topological interactions to understand the specific behaviours associated with various swarms [37,38]. Moreover, there are recent approaches by explicitly incorporating the sensory information such as visual sensing of the individuals or inferring the fine-scale social interaction rules among the individuals that help to get an insight into the coordinated motion of collectively moving groups [39–41]. However, it is quite challenging to construct the visual field and analyse the trajectories of every individual of large swarms in natural fields. Besides, the ways of communication in the groups may vary with changing environmental conditions. In such cases, these two broad groups of local interactions, namely metric and topological rules that indirectly incorporate the sensory cues and cognitive capabilities of individuals in a group, provide a great deal of information about the collective dynamics, as shown by our study. Our minimal model set-up could further be extended by incorporating other interactions as observed in nature, the effect of strong noise, or the presence of restrictive boundary or a predator attack and explore how it could influence the outcomes.

Data accessibility. Simulation codes to generate data of all figures in the paper and electronic supplementary materials are uploaded on Dryad. (Dryad Dataset, https://doi.org/10.5061/dryad.rjdfn2z91)
Authors' contributions. R.D. conceptualized the research. V.K. carried out the simulations. R.D. and V.K. analysed the data. V.K. wrote the first draft. R.D. wrote the manuscript and incorporated reviewers' suggestions.
Competing interests. The authors declare no competing financial interests.

Funding. The authors acknowledge the financial support from the Science and Engineering Research Board (SERB), grant no. SR/FTP/PS-105/2013, Department of Science and Technology (DST), India.

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
