## [Peer Review File · Royal Society Open Science]

Review History

RSOS-202158.R0 (Original submission)

Review form: Reviewer 1

Is the manuscript scientifically sound in its present form?

Yes

Are the interpretations and conclusions justified by the results?

No

Is the language acceptable?

Yes

Do you have any ethical concerns with this paper?

No

Have you any concerns about statistical analyses in this paper?

Yes

Recommendation?

Major revision is needed (please make suggestions in comments)

Comments to the Author(s)

See attachment (Appendix A).

Review form: Reviewer 2

Is the manuscript scientifically sound in its present form?

Yes

Are the interpretations and conclusions justified by the results?

Yes

Is the language acceptable?

Yes

Do you have any ethical concerns with this paper?

No

Have you any concerns about statistical analyses in this paper?

Yes

Recommendation?

Major revision is needed (please make suggestions in comments)

Comments to the Author(s)

See attached "Review.docx" file (Appendix B).

Decision letter (RSOS-202158.R0)

Dear Dr De

The Editors assigned to your paper RSOS-202158 "Efficient Flocking: Metric Versus Topological Interactions" have now received comments from reviewers and would like you to revise the paper in accordance with the reviewer comments and any comments from the Editors. Please note this decision does not guarantee eventual acceptance.

Please submit your revised manuscript and required files (see below) no later than 21 days from today's (ie 15-Mar-2021) date. Note: the ScholarOne system will 'lock' if submission of the revision is attempted 21 or more days after the deadline. If you do not think you will be able to meet this deadline please contact the editorial office immediately.

on behalf of Professor Roland Bouffanais (Associate Editor) and Miles Padgett (Subject Editor)
openscience@royalsociety.org

Associate Editor Comments to Author (Professor Roland Bouffanais):

Associate Editor: 1

Comments to the Author:

Two reviewers have come back to us with detailed reports. Although your work has some merit, major revisions are required to several parts of your manuscript. One review pointed out the omission of relevant references in your literature. review.

We hope that you would be able to address all the comments/issues raised by both reviewers in a revised version.

Associate Editor: 2

Comments to the Author:

(There are no comments.)

Reviewer comments to Author:

Reviewer: 1

Comments to the Author(s)

See attachment.

Reviewer: 2

Comments to the Author(s)

See attached "Review.docx" file

===PREPARING YOUR MANUSCRIPT===

===PREPARING YOUR REVISION IN SCHOLARONE===

- An individual file of each figure (EPS or print-quality PDF preferred [either format should be produced directly from original creation package], or original software format).
- An editable file of each table (.doc, .docx, .xls, .xlsx, or .csv).
- An editable file of all figure and table captions.

- Any electronic supplementary material (ESM).
- If you are requesting a discretionary waiver for the article processing charge, the waiver form must be included at this step.
- If you are providing image files for potential cover images, please upload these at this step, and inform the editorial office you have done so. You must hold the copyright to any image provided.
- A copy of your point-by-point response to referees and Editors. This will expedite the preparation of your proof.

- Ensure that your data access statement meets the requirements at <https://royalsociety.org/journals/authors/author-guidelines/#data>. You should ensure that you cite the dataset in your reference list. If you have deposited data etc in the Dryad repository, please include both the 'For publication' link and 'For review' link at this stage.
- If you are requesting an article processing charge waiver, you must select the relevant waiver option (if requesting a discretionary waiver, the form should have been uploaded at Step 3 'File upload' above).
- If you have uploaded ESM files, please ensure you follow the guidance at <https://royalsociety.org/journals/authors/author-guidelines/#supplementary-material> to include a suitable title and informative caption. An example of appropriate titling and captioning may be found at https://figshare.com/articles/Table_S2_from_Is_there_a_trade-off_between_peak_performance_and_performance_breadth_across_temperatures_for_aerobic_scope_in_teleost_fishes_/3843624.

Author's Response to Decision Letter for (RSOS-202158.R0)

See Appendices C-E.

RSOS-202158.R1 (Revision)

Review form: Reviewer 1

Is the manuscript scientifically sound in its present form?

Yes

Are the interpretations and conclusions justified by the results?

Yes

Is the language acceptable?

Yes

Do you have any ethical concerns with this paper?

No

Have you any concerns about statistical analyses in this paper?

No

Recommendation?

Accept with minor revision (please list in comments)

Comments to the Author(s)

I appreciate the authors' careful revision. The paper has been improved. For this version, just a couple of minor comments.

1. A log growth is likely for ϕ . Have you tried using log scale for the x-axis, i.e., T? Perhaps you could get a linear line.
2. In Figure 2, some parts of the curves are occluded by the legend. Please revise.

Review form: Reviewer 2

Is the manuscript scientifically sound in its present form?

Yes

Are the interpretations and conclusions justified by the results?

Yes

Is the language acceptable?

Yes

Do you have any ethical concerns with this paper?

No

Have you any concerns about statistical analyses in this paper?

Yes

Recommendation?

Accept with minor revision (please list in comments)

Comments to the Author(s)

See attached file (see Appendix F).

Decision letter (RSOS-202158.R1)

Dear Dr De,

On behalf of the Editors, we are pleased to inform you that your Manuscript RSOS-202158.R1 "Efficient Flocking: Metric Versus Topological Interactions" has been accepted for publication in Royal Society Open Science subject to minor revision in accordance with the referees' reports. Please find the referees' comments along with any feedback from the Editors below my signature.

Please submit your revised manuscript and required files (see below) no later than 7 days from today's (ie 02-Sep-2021) date. Note: the ScholarOne system will 'lock' if submission of the revision is attempted 7 or more days after the deadline. If you do not think you will be able to meet this deadline please contact the editorial office immediately.

on behalf of Professor Roland Bouffanais (Associate Editor) and Miles Padgett (Subject Editor)
openscience@royalsociety.org

Associate Editor Comments to Author (Professor Roland Bouffanais):

Your manuscript has been greatly improved during the previous round of revision. However, the Reviewers still recommend some minor revisions, which should easily be implemented. We look forward to receiving your second revision.

Reviewer comments to Author:

Reviewer: 1

Comments to the Author(s)

I appreciate the authors' careful revision. The paper has been improved. For this version, just a couple of minor comments.

1. A log growth is likely for ϕ . Have you tried using log scale for the x-axis, i.e., T? Perhaps you could get a linear line.
2. In Figure 2, some parts of the curves are occluded by the legend. Please revise.

Reviewer: 2
Comments to the Author(s)
See attached file ("**Review 2.pdf**").

===PREPARING YOUR MANUSCRIPT===

Your revised paper should include the changes requested by the referees and Editors of your manuscript. You should provide two versions of this manuscript and both versions must be provided in an editable format:
one version identifying all the changes that have been made (for instance, in coloured highlight, in bold text, or tracked changes);
a 'clean' version of the new manuscript that incorporates the changes made, but does not highlight them. This version will be used for typesetting.
Please ensure that any equations included in the paper are editable text and not embedded images.

===PREPARING YOUR REVISION IN SCHOLARONE===

At Step 3 'File upload' you should include the following files:
-- Your revised manuscript in editable file format (.doc, .docx, or .tex preferred). You should upload two versions:

-- If you have uploaded ESM files, please ensure you follow the guidance at <https://royalsociety.org/journals/authors/author-guidelines/#supplementary-material> to include a suitable title and informative caption. An example of appropriate titling and captioning may be found at [https://figshare.com/articles/Table_S2_from_Is_there_a_trade-off_between_peak_performance_and_performance_breadth_across_temperatures_for_aerobic_sc ope_in_teleost_fishes_/3843624](https://figshare.com/articles/Table_S2_from_Is_there_a_trade-off_between_peak_performance_and_performance_breadth_across_temperatures_for_aerobic_scope_in_teleost_fishes_/3843624).

Author's Response to Decision Letter for (RSOS-202158.R1)

See Appendix G-I.

Decision letter (RSOS-202158.R2)

Dear Dr De,

I am pleased to inform you that your manuscript entitled "Efficient Flocking: Metric Versus Topological Interactions" is now accepted for publication in Royal Society Open Science.

At present, we note that the email address for co-author "vk13ms149@iiserkol.ac.in" is currently marked as invalid. Please amend this within ScholarOne, or kindly reply to this email with the updated email address.

on behalf of Professor Roland Bouffanais (Associate Editor) and Miles Padgett (Subject Editor)
openscience@royalsociety.org

Appendix A

This paper studies the way the range of cooperative interaction versus the number of fixed interacting neighbours affects the dynamics of flocking in an unbounded space. It reveals the existence of a certain threshold value of the interaction radius in the case of metric ruling and a threshold number of interacting neighbours for the topological ruling to reach to an ordered state. It is shown that topological interaction is more effective in bringing the order in the group. The topic is worth investigation, and the results are interesting. However, there are some unclarities in the methodology and some relevant works are missing. Some specific comments are as follows.

1. In what space the velocity vector v_i in equation (1) is considered? If it's 2-d space, it would be better to give the components and make the definitions clearer. Moreover, if only 2-d space is considered, what would you expect for the real space, ie 3-d space?
2. The noise η_i in (1) is a linear noise. Why not consider a angular one? Any motivation for the choice? The normalization for V , the notations for α_0 and l_0 are confusing. You used i for the i -the particle. Therefore, readers then to believe the subscript 0 is $i=0$. However, this seems not to be the case. In general, I think the system (1) needs more justifications and remarks.
3. As the paper argues for the real world implications in flocking, the choice of number of particles N as 100-500 should be motivated.
4. What is δ in (3)? Is the unit velocity vector the same as dimensionless velocity? The definitions are missing. This is a common issue for many of the expressions in this paper.
5. In Figure 2, it is interesting that when $T=5$, the result is non-monotonic. Could you give some explanations for this phenomenon? Moreover, when r is greater than 28, there is a cross over phenomenon in both (a) and (b).
6. I suggest using log scale for the comparison between topological and metric vs N_r . The current qualitative analysis could be amended. In particular, in Figure 4, the discrepancy could be more prominent.
7. As the paper addresses metric vs topological interactions in flocking. The following two relevant works should be compared. [1] Influence of the number of topologically interacting neighbors on swarm dynamics; [2] Consensus reaching in swarms ruled by a hybrid metric-topological distance.
8. In Figure S2, are the results based on average or just one sample? The methodology is missing.

Appendix B

Review for

Efficient flocking: metric versus topological interactions

This manuscript describes a Vicsek-like self-propelled particle model where agents exhibit alignment with neighbours following either metric interactions (alignment with neighbours within a fixed radius), or topological interactions (alignment with a fixed number of nearest neighbours). The authors explore the effect of the interaction parameter (distance or number of nearest neighbours), speed, and size of the flock, on alignment of the group (measured here by the order parameter). They also define an efficiency metric; a group is considered more efficient if it achieves order by interacting with fewer individuals that are relatively close. Using this as the evaluation metric, the authors suggest topological interactions are more effective at bringing order than metric interactions.

The paper is overall clear and well-written, and the authors present their work in a coherent manner. Nonetheless, we have a few concerns we would like to raise.

Major Comments

One main concern with the manuscript is a lack of interpretation of the results from a biological perspective. The discussion section is short, and fails to highlight the relevance of these results for moving animal groups. We recommend that the authors either pitch this purely from a physics perspective to a different audience, or review the collective behaviour literature more thoroughly and re-interpret their results in the context of existing work in the field. We would like to see a more in-depth discussion on the relevance of this work for understanding the mechanisms of flocking. This leads us to our second major concern.

The study of collective animal behaviour has made significant progress over the past decade. Approaches to the study of inter-individual interactions have evolved and moved beyond simply considering metric and topological interactions (see Strandburg-Peshkin et al. 2013, Torney et al. 2018 and Bastien & Romanczuk 2020 for examples of theoretical and empirical work on this). With that in mind, it is unclear to us what new insights this particular work brings to the field. To be clear, we are not saying this work does not bring novel insights, just that the authors fail to discuss this in the context of existing work in the field.

Finally, more information should be included when reporting the model results, including appropriate quantification of error and number of simulations per parameter condition. Currently, some of the results are interpreted purely based on visual assessment of two different figures.

Minor Comments

In the abstract and multiple other places in the manuscript, the authors refer to these interactions as cooperative. There is little evidence suggesting flocking interactions are cooperative. Animals flock for varied reasons from improved information processing, environmental sensing, to predator avoidance. But none of these reasons indicate cooperation.

P.2, Column 1, L46–L49: ... alignment of velocities along with the nearest neighbours (Add reference)

P.2, Column 1, L51: Typo (full stop should probably be a comma)

P.2, Column 2, L30–L33, the authors state that it is unclear what type of interactions govern macroscopic collective behaviours in animal groups. However, this statement is at odds with the previous sentence.

P.3, Column 1, L7–L9: the implications (if any) of using an open space compared to periodic boundaries should be addressed in the discussion in the light of the model results, otherwise this doesn't need mentioning. In addition, same column, L46–L47, it is unclear why it is now referred to a square box of size L .

P.4, Column 1, L55–L56: this result seems rather counter-intuitive. We suggest the authors address why this trend emerges (both mathematically and biologically)

P.4, Column 2, L35: the authors refer to individuals as communicating among themselves. However, this is not true. A key feature of such schooling interactions is that it requires no explicit communication. These dynamics arise simply from individuals responding to the movements of groupmates.

Appendix C

To

Prof. Roland Bouffanais (Associate Editor) and Prof. Miles Padgett (Subject Editor),
Royal Society Open Science

Subject: Resubmission of Manuscript titled "Efficient Flocking: Metric Versus Topological Interactions" in Royal Society Open Science. Article Reference ID RSOS-202158

Dear Profs. Bouffanais and Padgett,

We thank you for giving us the opportunity to resubmit our manuscript to Royal Society Open Science. We also thank the reviewers for constructive suggestions that helped improve the quality of the manuscript to a great extent. We have explained all queries and incorporated all suggestions made by the reviewers in the revised manuscript.

Given the comments of the two reviewers, we have now revised our manuscript, and the major revisions that have been incorporated in our manuscript are highlighted in bold fonts. We have also addressed the questions raised by the reviewers point by point in the attached response letter to the reviewers.

We hope that these revisions would meet the approval of the reviewers. We thank you for your kind consideration of this manuscript.

Yours sincerely,

Rumi De

Associate Professor

Department of Physical Sciences

Indian Institute of Science Education and Research Kolkata

Mohanpur 741 246, West Bengal, India

Email: rumi.de@iiserkol.ac.in

Appendix D

Response to Reviewer 1:

First of all, we thank the reviewer for appreciative comments and constructive suggestions that helped improve the quality of the manuscript to a great extent. At the outset, we state that we have implemented all major suggestions in the revised manuscript. We now address the reviewer's questions point by point in the following.

This paper studies the way the range of cooperative interaction versus the number of fixed interacting neighbours affects the dynamics of flocking in an unbounded space. It reveals the existence of a certain threshold value of the interaction radius in the case of metric ruling and a threshold number of interacting neighbours for the topological ruling to reach to an ordered state. It is shown that topological interaction is more effective in bringing the order in the group. The topic is worth investigation, and the results are interesting. However, there are some unclarities in the methodology and some relevant works are missing. Some specific comments are as follows.

1. In what space the velocity vector \mathbf{v}_i in equation (1) is considered? If it's 2-d space, it would be better to give the components and make the definitions clearer. Moreover, if only 2-d space is considered, what would you expect for the real space, ie 3-d space?

• The velocity vector \mathbf{v}_i in equation (1) is considered in 2D space as $\mathbf{v}_i = v_0 \cos \theta_i \hat{x} + v_0 \sin \theta_i \hat{y}$, where v_0 is the magnitude and θ_i is the direction of the velocity of the i^{th} particle in the group. As suggested by the reviewer, we have now explicitly written the components to make it clearer in the revised manuscript.

We expect that the results will be qualitatively the same in 3D space; although, the optimal value of the threshold interaction radius in the case of metric interactions and the threshold number of interacting neighbours in topological interactions may vary. However, the explicit comparison is beyond the scope of the present manuscript.

2. The noise η_i in (1) is a linear noise. Why not consider a angular one? Any motivation for the choice? The normalization for \mathbf{V} , the notations for α_0 and l_0 are confusing. You used i for the i -the particle. Therefore, readers then to believe the subscript 0 is $i=0$. However, this seems not to be the case. In general, I think the system (1) needs more justifications and remarks.

• An angular noise could also be considered as done in several studies. However, since the main focus of our study is to investigate how the nature of the interactions, namely, the metric interactions versus the topological interactions, influence the flock; here we concentrate on the simple case where the noise is negligible.

As suggested by the reviewer, we have now changed the notation of the dimensionless variables, the normalization for \mathbf{V} , α_0 , and l_0 to avoid any confusion. Now, we have also explained each term of equation (1) in the revised manuscript.

3. As the paper argues for the real world implications in flocking, the choice of number of particles N as 100 – 500 should be motivated.

- The number is motivated from the literature study of the group size of flocks, schools, and herds of various species (for example, Couzin *et al.* have studied the groups, $N = 10$ to 200; Nature 433, 513–516, 2005 and the references therein). However, the many groups could also contain a large number of individuals. In our study, we have considered the group ranging from $N \sim 100 - 500$. Here, we would like to emphasize that even within this range, our analysis clearly demonstrates the influence of variation in group sizes in the case of both metric and topological interactions. Further, we have also shown the efficiency parameter, η , by normalizing the group size, N , to compare the trend for varying flock sizes to eliminate the dependence on a particular number. As the reviewer has suggested, we have now mentioned the motivation of the choice of the number of particles N in the revised manuscript.

4. What is delta in (3)? Is the unit velocity vector the same as dimensionless velocity? The definitions are missing. This is a common issue for many of the expressions in this paper.

- We thank the reviewer for pointing this out. In equation (3), the Kronecker delta, $\delta(r - r_{ij}) = 1$ if $r = r_{ij}$, and $\delta(r - r_{ij}) = 0$ if $r \neq r_{ij}$. We have now clearly mentioned it in the revised manuscript. Moreover, the unit velocity vector is not the same as dimensionless velocity. The unit vector is defined as, $\widehat{\mathbf{V}}_i = \frac{\mathbf{V}_i}{|\mathbf{V}_i|}$, where \mathbf{V}_i is the velocity of the i^{th} particle, and $|\mathbf{V}_i|$ is the magnitude, *i.e.*, the speed of the i^{th} particle. The dimensionless velocity is defined as $\mathbf{V} = \mathbf{v}/v_s$, where $v_s = l_s/\tau$ and the constants l_s and τ represent the characteristic length and time scale of the system. We have now clearly defined and discussed all expressions in the revised manuscript.

5. In Figure 2, it is interesting that when $T=5$, the result is non-monotonic. Could you give some explanations for this phenomenon? Moreover, when r is greater than 28, there is a cross over phenomenon in both (a) and (b).

- Figure 2 shows some representative plots of the velocity correlation function, $C(r)$, between a pair of particles, i and j , at a distance r as time progresses and the group approaches the flocking state (spatial average is calculated by summing over all such pairs for one simulation trajectory). As seen from Figure 2(a)-2(b), initially, since the particles are positioned randomly, thus, in the beginning at $T = 1$, the velocity correlation function is close to zero over the entire spatial extent r of the system. As time evolves, at $T = 5$, the particles which are located within the interaction radius $R = 5$ start interacting and gradually tend to align along with its neighbours. Thus, the correlation between the particle velocities increases at a shorter distance r ; however, the correlation between the particles at a larger distance, such as $r \sim 28$, stays close to zero. The crossover or the fluctuations at a large distance, r , where the value of $C(r)$ stays close to zero, shows the uncorrelated motion of the particles. However, as time progresses further, at $T = 1000$, the correlation functions, $C(r)$, approaches unity, and the entire group reaches an ordered state. We have now discussed it in detail in the revised manuscript.

6. I suggest using log scale for the comparison between topological and metric vs N_r . The current qualitative analysis could be amended. In particular, in Figure 4, the discrepancy could be more prominent.

- As the reviewer has suggested, we have checked using log scale and plotted Figure 4; however,

since the order parameter, ϕ , varies over a small range of values from $(0 - 1)$ and N_r from $(0 - 50)$, the discrepancy did not appear to be more prominent. If the values vary over a few orders of magnitudes, then the comparison using the log scale would have been more prominent. Therefore, we have kept the plot on a linear scale.

7. As the paper addresses metric vs topological interactions in flocking. The following two relevant works should be compared. [1] Influence of the number of topologically interacting neighbors on swarm dynamics; [2] Consensus reaching in swarms ruled by a hybrid metric-topological distance.

• We thank the reviewer. These are indeed relevant references to our study. We have now included these references and compared these studies in the revised manuscript. Changes have been highlighted in bold fonts in the revised manuscript.

8. In Figure S2, are the results based on average or just one sample? The methodology is missing.

• In Figure S2, we have calculated the spatial statistical average by summing over all pair of particles in a group as described by Eq 3 for one representative simulation trajectory. This gives a fair estimate of the velocity correlation, $C(r)$, between two individuals in a group at a distance r at a certain time instant T . We have now clearly described the methodology in the revised manuscript, and it is highlighted in bold fonts.

We hope that with these clarifications and revisions, this would now meet the approval of the reviewer.

Appendix E

Response to Reviewer 2:

Review for Efficient flocking: metric versus topological interactions

This manuscript describes a Vicsek-like self-propelled particle model where agents exhibit alignment with neighbours following either metric interactions (alignment with neighbours within a fixed radius), or topological interactions (alignment with a fixed number of nearest neighbours). The authors explore the effect of the interaction parameter (distance or number of nearest neighbours), speed, and size of the flock, on alignment of the group (measured here by the order parameter). They also define an efficiency metric; a group is considered more efficient if it achieves order by interacting with fewer individuals that are relatively close. Using this as the evaluation metric, the authors suggest topological interactions are more effective at bringing order than metric interactions.

The paper is overall clear and well-written, and the authors present their work in a coherent manner. Nonetheless, we have a few concerns we would like to raise.

First of all, we thank the reviewer for appreciative comments and constructive suggestions that helped improve the quality of the manuscript to a great extent. At the outset, we state that we have implemented all major suggestions in the revised manuscript. We now address the reviewer's questions point by point in the following.

Major Comments

One main concern with the manuscript is a lack of interpretation of the results from a biological perspective. The discussion section is short, and fails to highlight the relevance of these results for moving animal groups. We recommend that the authors either pitch this purely from a physics perspective to a different audience, or review the collective behaviour literature more thoroughly and re-interpret their results in the context of existing work in the field. We would like to see a more in-depth discussion on the relevance of this work for understanding the mechanisms of flocking.

- The fascinating display of collective behaviours has raised interest among biologists, physicists, and a diverse community of scientists working in interdisciplinary areas. We believe that our work will be of interest to the broad audience at the interface of physics and biology of this journal. As the reviewer has suggested, we have now reviewed the collective behaviour literature more thoroughly and highlighted the existing works in the context of our study on metric versus topological interactions. We have now rewritten the introduction section and also incorporated a more in-depth discussion in the revised manuscript.

The study of collective animal behaviour has made significant progress over the past decade. Approaches to the study of inter-individual interactions have evolved and moved beyond simply considering metric and topological interactions (see Strandburg-Peshkin et al. 2013, Torney et al. 2018 and Bastien & Romanczuk 2020 for examples of theoretical and empirical work on this). With that in mind, it is unclear to us what new insights this particular work brings to the field. To be clear, we are not saying this work does not bring novel insights, just that the authors fail to discuss this in

the context of existing work in the field.

- We agree with the reviewer that apart from these two broad groups of local interactions - metric and topological rules, there are also other recent approaches by explicitly incorporating sensory information such as visual sensing of the individuals or inferring the fine-scale social interaction rules among the individuals that help to get an insight into the coordinated motion of collectively moving groups (Strandburg-Peshkin et al. 2013, Torney et al. 2018 and Bastien & Romanczuk 2020). However, it is quite challenging to construct the visual field and analyse the trajectories of every individual in large swarms in natural fields. Besides, the ways of communication in the groups may vary with changing environmental conditions. In such cases, these two broad groups of local interactions, namely metric and topological rules that indirectly incorporate the sensory cues and cognitive capabilities of individuals in a group, provide a great deal of information about the collective dynamics as shown by our study and also other works. We have now included a discussion on this in the revised manuscript.

Finally, more information should be included when reporting the model results, including appropriate quantification of error and number of simulations per parameter condition. Currently, some of the results are interpreted purely based on visual assessment of two different figures.

- As the reviewer has suggested, we have now included the information on number of simulations per parameter condition. The results are obtained averaging over hundred such simulation trajectories starting from different initial configurations. Moreover, apart from the visual assessment of the plots, we have also calculated the exact values of the threshold interaction radius and the number of interacting neighbours in case of both metric and topological interactions for various flock sizes and speeds to interpret the results, and the values are presented in Table 1 in the supplementary section.

Minor Comments

In the abstract and multiple other places in the manuscript, the authors refer to these interactions as cooperative. There is little evidence suggesting flocking interactions are cooperative. Animals flock for varied reasons from improved information processing, environmental sensing, to predator avoidance. But none of these reasons indicate cooperation.

- As the reviewer has pointed out, we have now omitted the word cooperative in the revised manuscript.

P.2, Column 1, L46–L49: ... alignment of velocities along with the nearest neighbours (Add reference)

- We have added the references.

P.2, Column 1, L51: Typo (full stop should probably be a comma)

- It has now been corrected.

P.2, Column 2, L30–L33, the authors state that it is unclear what type of interactions govern macroscopic collective behaviours in animal groups. However, this statement is at odds with the previous sentence.

- As suggested by the reviewer, we have now modified these sentences in the revised manuscript.

P.3, Column 1, L7–L9: the implications (if any) of using an open space compared to periodic boundaries should be addressed in the discussion in the light of the model results, otherwise this doesn't need mentioning. In addition, same column, L46–L47, it is unclear why it is now referred to a square box of size L .

- We have considered that the particles move in open space; hence, the boundary is not restricted. We have mentioned it since the boundary condition is required to solve the equation of motion of the particles. As the reviewer has suggested, we have now omitted the sentence on periodic boundary condition in the revised manuscript.

Also, to state the initial condition, we have referred to a square box of size L . In our simulations, we have considered a group of N particles initially positioned randomly in a square box of size L , and the positions and velocities of the particles evolve in time following Equation (1). We have now clearly mentioned it in the revised manuscript.

P.4, Column 1, L55–L56: this result seems rather counter-intuitive. We suggest the authors address why this trend emerges (both mathematically and biologically)

- As suggested by the reviewer, we have now addressed and explained this trend in the revised manuscript.

P.4, Column 2, L35: the authors refer to individuals as communicating among themselves. However, this is not true. A key feature of such schooling interactions is that it requires no explicit communication. These dynamics arise simply from individuals responding to the movements of groupmates.

- We agree with the reviewer. As suggested, we have now modified the sentence in the revised manuscript.

We hope that with these clarifications and revisions, this would now meet the approval of the reviewer.

Appendix F

Review for

Efficient flocking: metric versus topological interactions

We thank the authors for their efforts in addressing our comments. They have worked on most of our concerns regarding the manuscript and we are happy with the revised version. Below we point out a few minor points that stood out.

To address our previous concern regarding quantification and reporting of error in the simulations, the authors have now added to the text that most plots result from averaging over hundred simulated trajectories (except figure 1 where it says it shows a representative simulation trajectory). In all cases, we would like the authors to report 95% confidence intervals around the mean (and plot them on the figures). This will also give the reader a sense of variability between replicate simulation runs.

P.2, Column 2, L35–L37: Typo (there are also other **metric** based)

P.3, Column 1, L10–L15: Sentence structure uses two “linking words of contrast”—although and however. Reword to use only one of the two

P.4, Column 1, L50: Typo (To calculate the **spatial** average)

Appendix G

Response to Reviewer 1:

I appreciate the authors' careful revision. The paper has been improved. For this version, just a couple of minor comments.

We thank the reviewer for the appreciative comments and for accepting the revised manuscript. We now address the reviewer's comments in the following.

1. A log growth is likely for phi. Have you tried using log scale for the x-axis, i.e., T? Perhaps you could get a linear line.

As the reviewer has suggested, we have tried plotting the x-axis, T , in log scale. As shown here, Figures 1(a) and 1(b) have been plotted in log-log scale and also in semi-log scale. Since the plots in log scale do not change the reported conclusion, we have kept the figures in linear scale in the manuscript.

Figure 1: **Figures 1(a) and 1(b) in log-log scale.** Order parameter, ϕ , as function of time, T , for different interaction radius ($R = 1, 5, 10$) and initial flock speed, V_0 , (a) at a lower speed ($V_0 = 0.01$) and (b) at a higher speed ($V_0 = 1.0$).

Figure 2: **Figures 1(a) and 1(b) in semi-log scale.** Order parameter, ϕ , as function of time, T , for different interaction radius ($R = 1, 5, 10$) and initial flock speed, V_0 , (a) at a lower speed ($V_0 = 0.01$) and (b) at a higher speed ($V_0 = 1.0$).

2. In Figure 2, some parts of the curves are occluded by the legend. Please revise.

We thank the reviewer for pointing this out. Figure legends were indeed overlapping some parts of the curves. We have now corrected the figures in the revised manuscript.

Appendix H

Response to Reviewer 2:

We thank the authors for their efforts in addressing our comments. They have worked on most of our concerns regarding the manuscript and we are happy with the revised version. Below we point out a few minor points that stood out.

We thank the reviewer for the appreciative comments and for accepting the revised manuscript. We now address the reviewer's comments in the following.

To address our previous concern regarding quantification and reporting of error in the simulations, the authors have now added to the text that most plots result from averaging over hundred simulated trajectories (except figure 1 where it says it shows a representative simulation trajectory). In all cases, we would like the authors to report 95% confidence intervals around the mean (and plot them on the figures). This will also give the reader a sense of variability between replicate simulation runs.

As we have written in the manuscript, Figure 1 and Figure 2 show a representative simulation trajectory, and for other figures, averaging has been done over hundred simulation trajectories. The steady-state values for different trajectories turn out to be almost the same since the simulations have been carried out in the absence of noise, as stated in the manuscript. In the case of simulation trajectories in the presence of noise, error bar plots would have been meaningful. Moreover, since errors bar plots would overlap with the curves making it difficult to distinguish the curves, therefore we have not plotted them.

P.2, Column 2, L35–L37: Typo (there are also other metric based)

P.3, Column 1, L10–L15: Sentence structure uses two “linking words of contrast”—although and however. Reword to use only one of the two

P.4, Column 1, L50: Typo (To calculate the spatial average)

We thank the reviewer for pointing out these mistakes. We have now corrected them in the revised manuscript.

Appendix I

To
Prof. Roland Bouffanais (Associate Editor) and Prof. Miles Padgett (Subject Editor),
Royal Society Open Science

Subject: Second revision of manuscript titled "Efficient Flocking: Metric Versus Topological Interactions" in Royal Society Open Science. Article Reference ID RSOS-202158

Dear Profs. Bouffanais and Padgett,

We thank you for accepting our revised manuscript. We also thank the reviewers for their appreciative comments and for accepting the revised manuscript. We have now incorporated most of the minor suggestions given by the reviewers in the second revision of the manuscript.

We hope that with these revisions, the manuscript will be accepted for publication in Royal Society Open Science.

Yours sincerely,
Rumi De
Associate Professor
Department of Physical Sciences
Indian Institute of Science Education and Research Kolkata
Mohanpur 741 246, West Bengal, India
Email: rumi.de@iiserkol.ac.in